# Towards Practical Few-Shot Query Sets: Transductive Minimum Description Length Inference

**Ségolène Martin**
Université Paris-Saclay, Inria,
CentraleSupélec, CVN

**Malik Boudiaf**
ÉTS Montreal

**Emilie Chouzenoux**[*]
Université Paris-Saclay, Inria,
CentraleSupélec, CVN

**Jean-Christophe Pesquet**[†]
Université Paris-Saclay, Inria,
CentraleSupélec, CVN

**Ismail Ben Ayed**[‡]
ÉTS Montreal

## Abstract

Standard few-shot benchmarks are often built upon simplifying assumptions on the query sets, which may not always hold in practice. In particular, for each task at testing time, the classes effectively present in the unlabeled query set are known a priori, and correspond exactly to the set of classes represented in the labeled support set. We relax these assumptions and extend current benchmarks, so that the query-set classes of a given task are unknown, but just belong to a much larger set of possible classes. Our setting could be viewed as an instance of the challenging yet practical problem of extremely imbalanced $K$-way classification, $K$ being much larger than the values typically used in standard benchmarks, and with potentially irrelevant supervision from the support set. Expectedly, our setting incurs drops in the performances of state-of-the-art methods. Motivated by these observations, we introduce a **P**rim**Al D**ual Minimum **D**escription **LE**ngth (**PADDLE**) formulation, which balances data-fitting accuracy and model complexity for a given few-shot task, under supervision constraints from the support set. Our constrained MDL-like objective promotes competition among a large set of possible classes, preserving only effective classes that befit better the data of a few-shot task. It is hyper-parameter free, and could be applied on top of any base-class training. Furthermore, we derive a fast block coordinate descent algorithm for optimizing our objective, with convergence guarantee, and a linear computational complexity at each iteration. Comprehensive experiments over the standard few-shot datasets and the more realistic and challenging *i-Nat* dataset show highly competitive performances of our method, more so when the numbers of possible classes in the tasks increase. Our code is publicly available at `https://github.com/SegoleneMartin/PADDLE`.

## 1   Introduction

The performance of deep learning models is often seriously affected when tackling new tasks with limited supervision, i.e., classes that were unobserved during training and for which we have only a handful of labeled examples. Few-shot learning [1, 2, 3] focuses on this generalization challenge, which occurs in a breadth of applications. In standard few-shot settings, a deep network is first trained

[*]E. Chouzenoux acknowledges support from the European Research Council Starting Grant MAJORIS ERC-2019-STG-850925.

[†]The work of J.-C. Pesquet is supported by the ANR Chair in AI BRIDGEABLE.

[‡]The work of I. Ben Ayed is supported by the DATAIA Institute, and is part of his sabbatical-leave visit at the Université Paris-Saclay.

on a large-scale dataset including labeled instances sampled from an initial set of classes, commonly referred to as the base classes. Subsequently, for novel classes, unobserved during the base training, supervision is restricted to a limited number of labeled instances per class. During the test phase, few-shot methods are evaluated over individual tasks, each including a small batch of unlabeled test samples (*the query set*) and a few labeled instances per novel class (*the support set*).

The transduction approach has becomes increasingly popular in few-shot learning, and a large body of recent methods focused on this setting, including, for instance, those based on graph regularization [4, 5], optimal transport [6, 7], feature transformations [8, 9], information maximization [10, 11, 12] and transductive batch normalization[13, 2], among other works [14, 15, 9, 16, 17]. Transductive few-shot classifiers make joint predictions for the batch of query samples of each few-shot task, independently of the other tasks. Unlike inductive inference, in which prediction is made for one testing sample at a time, transduction accounts for the statistics of the query set of a task[4], typically yielding substantial improvements in performance. On standard benchmarks, the gap in classification accuracy between transductive and inductive few-shot methods may reach $10\%$; see [11], for example. This connects with a well-known fact in classical literature on transductive inference [19, 20, 21], which prescribes transduction as an effective way to mitigate the difficulty inherent to learning from limited labels. It is worth mentioning that transductive methods inherently depend on the statistical properties of the query sets. For instance, the recent studies in [22, 10] showed that variations in the class balance within the query set may affect the performances of transductive methods.

Transductive few-shot classification occurs in a variety of practical scenarios, in which we naturally have access to a batch of unlabeled samples at test time. In commonly used few-shot benchmarks, the query set of each task is small (less than 100 examples), and is sampled from a limited number of classes (typically 5). Those choices are relevant in practice: During test time, one typically has access to small unlabeled batches of potentially correlated (non-i.i.d.) samples, e.g., smart-device photos taken at a given time, video-stream sequences, or in pixel prediction tasks such as semantic segmentation [12], where only a handful of classes appear in the testing batch. However, the standard few-shot benchmarks are built upon further assumptions that may not always hold in practice: *the few classes effectively present in the unlabeled query set are assumed both to be known beforehand and to match exactly the set of classes of the labeled support set.* We relax these assumptions, and extend current benchmarks so that the query-set classes are unknown and do not match exactly the support-set classes, but just belong to a much larger set of possible classes. Specifically, we allow the total number of classes represented in the support set to be higher than typical values, while keeping the number of classes that are effectively present in the query set to be much smaller.

Our challenging yet practical setting raises several difficulties for state-of-the-art transductive few-shot classifiers: (i) it is an instance of the problem of highly imbalanced classification; (ii) the labeled support set includes "distraction" classes that may not actually be present in the test query samples; (iii) we consider $K$-way classification tasks with $K$ much larger than the typical values used in the current benchmarks. We evaluated 7 of the best-performing state-of-the-art transductive few-shot methods and, expectedly, observed drops in their performance in this challenging setting.

Motivated by these experimental observations, we introduce a Minimum Description Length (MDL) inference, which balances data-fitting accuracy and model complexity for a given few-shot task, subject to supervision constraints from the support set. The model-complexity term can be viewed as a continuous relaxation of a discrete label cost, which penalizes the number of non-empty clusters in the solution, fitting the data of a given task with as few unique labels as necessary. It encourages competition among a large set of possible classes, preserving only those that fit better the task. Our formulation is hyper-parameter free, and could be applied on top of any base-class training. Furthermore, we derive a fast primal-dual block coordinate descent algorithm for optimizing our objective, with convergence guarantee, and a linear computational complexity at each iteration thanks to closed-form updates of the variables. We report comprehensive experiments and ablation studies on *mini*-Imagenet, *tiered*-Imagenet, and the more realistic and challenging iNat dataset [23] for fine-grained classification, with 908 classes and 227 ways at test-time. Our method yields competitive performances in comparison to the state-of-the-art, with gaps increasing substantially with the numbers of possible classes in the tasks.

---

[4]Note that the transductive setting is different from semi-supervised few-shot learning [18] that uses extra data. The only difference between inductive and tranductive inference is that predictions are made jointly for the query set of a task, rather than one sample at a time.

## 2 Few-shot setting and task generation

**Base training** Let $\mathcal{D}_{\text{base}} = \{\boldsymbol{x}_n, \boldsymbol{y}_n\}_{n=1}^{|\mathcal{D}_{\text{base}}|}$ denotes the base dataset, where each $\boldsymbol{x}_n \in \mathcal{X}_{\text{base}}$ is a sample from some input space $\mathcal{X}$, $\boldsymbol{y}_n \in \{0,1\}^{|\mathcal{Y}_{\text{base}}|}$ the associated ground-truth label, and $\mathcal{Y}_{\text{base}}$ the set of base classes. Base training learns a feature extractor $f_\phi : \mathcal{X} \to \mathcal{Z}$, with parameters $\phi$ and $\mathcal{Z}$ a lower-dimensional space. For this stage, an abundant few-shot literature adopts episodic training, which views $\mathcal{D}_{\text{base}}$ as a series of tasks (or episodes) so as to simulate testing time. Then, a meta-learner is devised to produce the predictions. However, it has been widely established over recent years that a basic training, followed by transfer-learning strategies, outperforms most meta-learning methods [24, 25, 26, 4, 11]. Hence, we adopt a standard cross-entropy training in this work.

**Evaluation** Evaluation is carried out over few-shot tasks, each containing samples from $\mathcal{D}_{\text{test}} = \{\boldsymbol{x}_n, \boldsymbol{y}_n\}_{n=1}^{|\mathcal{D}_{\text{test}}|}$, where $\boldsymbol{y}_n \in \{0,1\}^{|\mathcal{Y}_{\text{test}}|}$, with constraint $\mathcal{Y}_{\text{base}} \cap \mathcal{Y}_{\text{test}} = \varnothing$, i.e., the test and base classes are distinct. Each task includes a labelled support set $\mathbb{S} = \{\boldsymbol{x}_n, \boldsymbol{y}_n\}_{n \in \mathbb{I}_\mathbb{S}}$ and an unlabelled query set $\mathbb{Q} = \{\boldsymbol{x}_n\}_{n \in \mathbb{I}_\mathbb{Q}}$, both containing examples from classes in $\mathcal{Y}_{\text{test}}$. Using the feature extractor $f_\phi$ trained on base data, the goal is to predict the classes of the unlabeled samples in $\mathbb{Q}$ for each few-shot task, independently of the other tasks.

**Task generation** For a given task, let $K$ denote the total number of possible classes that are represented in the labeled support set $\mathbb{S}$, and $K_{\text{eff}}$ the number of classes that appear effectively in unlabeled query set $\mathbb{Q}$ (i.e. classes represented by at least one sample). The standard task generation protocol assumes that the set of effective classes in $\mathbb{Q}$ matches exactly the set of classes in $\mathbb{S}$, i.e. $K = K_{\text{eff}} \ll |\mathcal{Y}_{\text{test}}|$, which amounts to knowing exactly the few classes that appear in the test samples. Then, a fixed number of instances per class are sampled for each query set, forcing class balance. We relax these assumptions, and propose a setting where the set of effective classes in $\mathbb{Q}$ is not known exactly and belongs to a much larger set of possible classes. First, we allow the total number of possible classes $K$ to be higher than typical values, with $K = |\mathcal{Y}_{\text{test}}|$, while keeping the number of effective classes in the query set to be typically much smaller: $K_{\text{eff}} \ll |\mathcal{Y}_{\text{test}}| = K$. For instance, in our experiments, $K$ can go up to 227. This, as expected, results in $K$-way problems that are more challenging than the standard 5-way tasks used in the few-shot literature. Secondly, once the effective classes are fixed, and given a budget of images, we sample the query set under the data joint distribution (i.e. uniform draws among all available examples), so that the statistics of the sampled query sets more faithfully reflect the natural distribution of classes. Figure 1 illustrates our framework (base training, inference and task generation).

## 3 Proposed few-shot inference formulation

For a given few-shot task, let $\mathbb{Q} = \{\boldsymbol{x}_n\}_{n \in \mathbb{I}_\mathbb{Q}}$ and $\mathbb{S} = \{\boldsymbol{x}_n, \boldsymbol{y}_n\}_{n \in \mathbb{I}_\mathbb{S}}$ be two subsets of $\mathcal{D}_{\text{test}}$ such that $\mathbb{Q} \cap \mathbb{S} = \varnothing$. Let $N = |\mathbb{Q}| + |\mathbb{S}|$. Up to a reordering, we can suppose that $\mathbb{I}_\mathbb{Q} = \{1, \dots, |\mathbb{Q}|\}$ and $\mathbb{I}_\mathbb{S} = \{|\mathbb{Q}| + 1, \dots, N\}$. We denote $\boldsymbol{z}_n = f_\phi(\boldsymbol{x}_n)$ the feature vector corresponding to the data sample $\boldsymbol{x}_n$, and $\boldsymbol{u}_n = (u_{n,k})_{1 \le k \le K} \in \{0,1\}^K$ the variable assigning the data to one of the possible classes in $\{1, \dots, K\}$, i.e. $u_{n,k} = 1$ if $\boldsymbol{x}_n$ belongs to class $k$ and 0 otherwise. We define the variables giving the class proportions $\hat{\boldsymbol{u}} = (\hat{u}_k)_{1 \le k \le K} \in \Delta_K$ as

$$\hat{u}_k = \frac{1}{|\mathbb{Q}|} \sum_{n=1}^{|\mathbb{Q}|} u_{n,k} \quad \forall k \in \{1, \dots, K\}, \tag{1}$$

where $\Delta_K$ is the unit simplex of $\mathbb{R}^K$.

We propose to cast transductive few-shot inference as the minimization of an objective balancing data-fitting accuracy and partition complexity, subject to supervision constraints (known labels) from the support set $\mathbb{S}$. Our approach estimates jointly $\boldsymbol{U} = (\boldsymbol{u}_n)_{1 \le n \le N}$, which defines a partition of

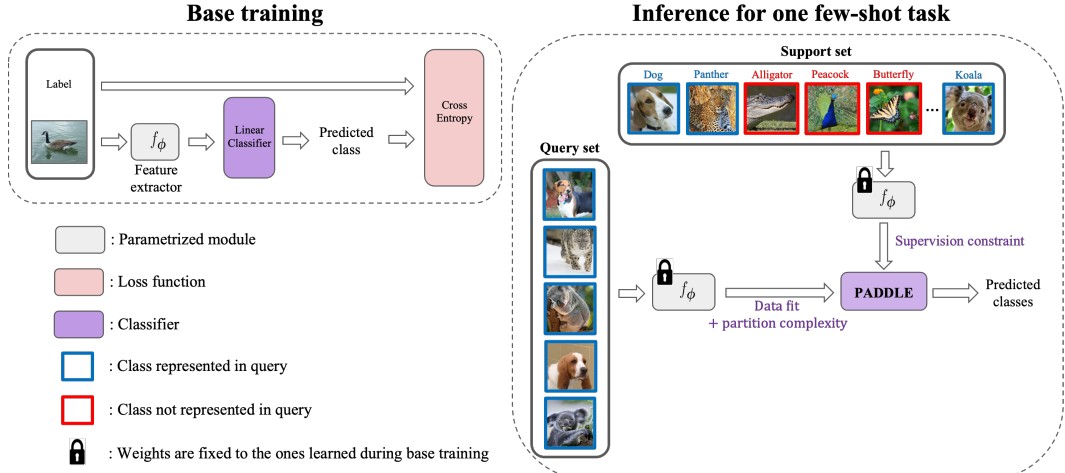

Figure 1: Overview of the proposed framework: Base training, PADDLE inference and task generation. The example depicted in the right-hand side illustrates how the support and query classes do not match exactly, unlike in standard few-shot settings. The support set includes "distraction" classes that may not actually be present in the query set, e.g. classes "Alligator", "Peacock" and "Butterfly". All possible classes ($K$ classes) are represented in the support set, but only an unknown subset among these $K$ possible classes ($K_{\text{eff}}$ effective classes) appear in the query set, with $K_{\text{eff}} \ll K$.

$\mathbb{S} \cup \mathbb{Q}$, and the class prototypes $\boldsymbol{W} = (\boldsymbol{w}_k)_{1 \leq k \leq K} \in (\mathbb{R}^d)^K$ through the following problem:

$$\underset{\boldsymbol{U}, \boldsymbol{W}}{\text{minimize}} \quad \underbrace{\frac{1}{2} \sum_{k=1}^{K} \sum_{n=1}^{N} u_{n,k} \|\boldsymbol{w}_k - \boldsymbol{z}_n\|^2}_{\text{data-fitting accuracy}} - \lambda \underbrace{\sum_{k=1}^{K} \hat{u}_k \ln(\hat{u}_k)}_{\text{partition complexity}}, \quad (2)$$

$$\text{s.t} \quad \boldsymbol{u}_n \in \Delta_K \quad \forall n \in \{1, \ldots, |\mathbb{Q}|\},$$
$$\boldsymbol{u}_n = \boldsymbol{y}_n \quad \forall n \in \{|\mathbb{Q}| + 1, \ldots, N\}. \quad (\text{C})$$

Note that, in the second line of (2), we have relaxed the integer constraints on query assignments to facilitate optimization.

**Effect of each term**   The purpose of objective (2) is to classify the data of a few-shot task with as few unique labels as necessary. The data-fitting term has the same form as the standard $K$-means objective for clustering, but is constrained with supervision from the support-set labels. This term evaluates, within each class $k$, the deviation of features from class prototype $\boldsymbol{w}_k$, thereby encouraging consistency of samples of the same class. The partition-complexity term implicitly penalizes the number of effective (non-empty) classes that appear in the solution ($K_{\text{eff}} \leq K$), encouraging low cardinality partitions of query set $\mathbb{Q}$. This term is the Shannon entropy of class proportions within $\mathbb{Q}$: it reaches its minimum when all the samples of $\mathbb{Q}$ belong to a single class, i.e., $\exists j \in \{1, \ldots, K\}$ such that $\hat{u}_j = 1$ and all other proportions vanish. It achieves its maximum for perfectly balanced partitions of $\mathbb{Q}$, satisfying $\hat{u}_k = 1/K$ for all $k$. In practice, this terms promotes solutions that contain only a handful of effective classes among a larger set of $K$ possible classes.

**Connection to Minimum Description Length (MDL)**   The objective in (2) could be viewed as a partially-supervised instantiation of the general MDL principle. Originated in information theory, MDL is widely used in statistical model selection [27]. Assume that we want to describe some input data $\boldsymbol{Z} = (\boldsymbol{z}_n)_{1 \leq n \leq |\mathbb{Q}|}$ with a statistical model $\boldsymbol{M}$, among a family of possible models. MDL prescribes that the best model corresponds to the *shortest* description of the data, according to some coding scheme underlying the model. It balances data-fitting accuracy and model complexity by minimizing $\mathcal{L}(\boldsymbol{Z}|\boldsymbol{M}) + \lambda \mathcal{L}(\boldsymbol{M})$ w.r.t $\boldsymbol{M}$. $\mathcal{L}(\boldsymbol{Z}|\boldsymbol{M})$ measures the *code length* of the prediction of $\boldsymbol{Z}$ made by $\boldsymbol{M}$, while $\mathcal{L}(\boldsymbol{M})$ encourages simpler models (Occam's razor [27]), e.g., models with less parameters. For instance, in unsupervised clustering, it is common to minimize a discrete label cost as a measure of model complexity [28, 29, 30]. Such a label cost is the number of effective (non-empty)

clusters in the solution, and is used in conjunction with a log-likelihood term for data-fitting:

$$\mathcal{L}(\boldsymbol{M}) = \sum_{k=1}^{K} 1_{\widehat{u}_k \neq 0} \tag{3}$$

$$\mathcal{L}(\boldsymbol{Z}|\boldsymbol{M}) = -\sum_{k=1}^{K} \sum_{n=1}^{|\mathbb{Q}|} u_{n,k} \ln \Pr(\boldsymbol{z}_n|k; \boldsymbol{w}_k) \tag{4}$$

In this expression, $\boldsymbol{M} = \{(\boldsymbol{w}_k)_{1 \leq k \leq K}, \boldsymbol{U}\}$ with $\boldsymbol{w}_k$ being the parameters of some probability distribution $\Pr(\boldsymbol{z}_n|k; \boldsymbol{w}_k)$ describing the samples in cluster $k$, while $\boldsymbol{U}$ contains assignments as denoted above.

The discrete model complexity in (3) is commonly used in MDL-based clustering [28, 29, 30], despite the ensuing optimization difficulty. As this measure is a discrete count of non-empty clusters, it does not accommodate fast optimization techniques. Typically, it is either handled via cluster merging heuristics (starting from an initial set of clusters) [30] or via combinatorial move-making algorithms [28], both of which are computationally intensive, more so when dealing with large sets of classes.

Our partition-complexity term in (2) can be viewed as a *continuous relaxation* of the discrete label count in (3) (in the supplemental material, we provide a graphical illustration in the case when $K = 2$). While penalizing similarly the number of effective classes, it has several advantages over the label count in (3), despite the surprising fact that it is not common in the MDL-based clustering literature, to our best knowledge. First, it is a continuous function of assignment variables $\boldsymbol{U}$. This enables us to derive a fast primal-dual block coordinate descent algorithm (Section 4), with convergence guarantee, and a linear computational complexity at each iteration thanks to closed-form updates of the variables. Second, it has a clear MDL interpretation: it measures the number of bits required to encode the set of classes, given class probabilities $\Pr(k) = \hat{u}_k, \forall k \in \{1, \ldots, K\}$.

Following the Kraft-McMillan theorem [27], any probability distribution $\Pr(\boldsymbol{z}_n|k; \boldsymbol{w}_k)$ corresponds to some coding scheme for storing the features of cluster $k$, and $-\ln \Pr(\boldsymbol{z}_n|k; \boldsymbol{w}_k)$ is the number of bits required to represent any feature using coding scheme $\Pr(\boldsymbol{z}_n|k; \boldsymbol{w}_k)$. Clearly, our data-fitting term in (2) also fits into this MDL interpretation by assuming that each probability $\Pr(\boldsymbol{z}_n|k; \boldsymbol{w}_k)$ is a Gaussian distribution with mean $\boldsymbol{w}_k$ and covariance fixed to the identity matrix:

$$\Pr(\boldsymbol{z}_n|k; \boldsymbol{w}_k) \propto \exp\left(-\frac{1}{2}\|\boldsymbol{w}_k - \boldsymbol{z}_n\|^2\right) \quad \forall k \in \{1, \ldots, K\}. \tag{5}$$

**Differences with the existing objectives for transductive few-shot inference**   Unlike most of the existing transductive few-shot methods, our MDL formulation does not encode strong assumptions on the label statistics of the query set (e.g. class balance or/and a perfect match between the query and support classes). Information maximization methods, such as [10, 11], maximise the confidence of the sample-wise predictions while encouraging class balance. Optimal transport methods [6, 7] estimate an optimal mapping matrix, which could be viewed as a joint probability distribution over the features and the labels, while imposing a hard class-balance constraint via the Sinkhorn-Knopp algorithm. Both information-maximization and optimal-transport methods have inherent class-balance bias and, as shown in the experiments, undergo a drastic drop in accuracy when the number of possible classes increases (as this corresponds to highly imbalanced problems). Prototype rectification [8] transforms the query features so as to minimize the difference between the overall statistics of the query and support sets. This relies on the assumption that the support and query classes match perfectly. Therefore, it is not adapted to our setting. Inspired from graphical models, the Laplacian regularization in [4] is a pairwise label correction (rather than a learning) method, which encourages assigning the same label to query samples that are close in the input space; it does not learn a class representation from the query set (the prototypes being fixed as those of the support set).

**Identifying $\lambda$ through an unbiased probabilistic K-means interpretation of** (2)   Another interesting view of our few-shot inference objective in (2) can be drawn from the probabilistic $K$-means objective[5], well-known in the clustering literature [31, 32, 33]. In fact, probabilistic $K$-means corresponds to minimizing $\mathcal{L}(\boldsymbol{Z}|\boldsymbol{M})$ in (4) as a generalization of $K$-means, which corresponds to the particular Gaussian choice in (5). It is a well-known that probabilistic $K$-means has a strong bias

---

[5]In the general context of clustering, the name probabilistic $K$-means was first coined by [31].

towards balanced partitions [31, 33]. The objective can be decomposed to reveal a hidden term promoting class balance. Using Bayes rule $\Pr(k|\boldsymbol{z}_n; \boldsymbol{w}_k) \propto \Pr(\boldsymbol{z}_n|k; \boldsymbol{w}_k)\Pr(k)$, empirical estimates of marginal class probabilities $\Pr(k) = \hat{u}_k$, and the choice of $\Pr(\boldsymbol{z}_n|k; \boldsymbol{w}_k)$ in (5), we have:

$$
\frac{1}{2}\sum_{k=1}^{K}\sum_{n=1}^{|\mathbb{Q}|} u_{n,k}\|\boldsymbol{w}_k - \boldsymbol{z}_n\|^2 \;=\; -\sum_{k=1}^{K}\sum_{n=1}^{|\mathbb{Q}|} u_{n,k}\ln\Pr(\boldsymbol{z}_n|k; \boldsymbol{w}_k),
$$

$$
\overset{\mathrm{c}}{=}\; -\sum_{k=1}^{K}\sum_{n=1}^{|\mathbb{Q}|} u_{n,k}\ln\Pr(k|\boldsymbol{z}_n; \boldsymbol{w}_k) + |\mathbb{Q}|\sum_{k=1}^{K}\hat{u}_k\ln(\hat{u}_k), \quad (6)
$$

where $\overset{\mathrm{c}}{=}$ stands for equality, up to an additive constant independent of optimization variables. Minimizing the last term in the second line of (6) has an effect opposite to the model-complexity term in our objective in (2): it reaches its minimum for perfectly balanced partitions. Therefore, our objective in (2) is a way to mitigate such a bias in $K$-means, allowing imbalanced partitions. This suggests setting $\lambda = |\mathbb{Q}|$ to compensate for the hidden class-balance term in $K$-means. This would make our inference hyper-parameter free. Thus, for our study, we fixed $\lambda = |\mathbb{Q}|$ in all benchmarks, without optimizing the parameter $\lambda$ via validation.

## 4 Primal-Dual Block-Coordinate Descent Optimization

We derive a block coordinate descent algorithm for our formulation in (2). At each iteration of our algorithm, the minimization steps are closed-form, with a linear complexity in $K$ and $N$. We emphasize that, in problem (2), the minimization over $\boldsymbol{U}$ could not be carried in closed-form, even for fixed prototypes $\boldsymbol{W}$, and the simplex constraints are difficult to handle. One straightforward iterative solution to tackle (2) would be to deploy a projected gradient descent algorithm. However, as shown in the comparisons in Section 5, this strategy is computationally demanding and does not yield better classification results than the algorithm we propose below.

**Primal-Dual formulation** We transform problem (2) into an equivalent optimization problem by introducing a dual variable. Let us first define the linear operator $\boldsymbol{A}\colon (\mathbb{R}^K)^N \longrightarrow \mathbb{R}^K$, which maps the probabilities to the proportions of the classes, i.e. $\boldsymbol{A} : \boldsymbol{U} \mapsto (\hat{u}_k)_{1\leq k\leq K}$. Then, problem (2) can be re-written as follows:

$$
\underset{\boldsymbol{U},\boldsymbol{W}}{\text{minimize}} \quad \frac{1}{2}\sum_{k=1}^{K}\sum_{n=1}^{N} u_{n,k}\|\boldsymbol{w}_k - \boldsymbol{z}_n\|^2 - \lambda H(\boldsymbol{A}\boldsymbol{U}), \quad \text{s.t} \quad (\mathrm{C})
$$

where $H$ is the negative entropy function on the positive orthant of $\mathbb{R}^K$: $H(\boldsymbol{x}) = \sum_{k=1}^{K}\varphi(x_k)$ for $\boldsymbol{x} = (x_k)_{1\leq k\leq K} \in \mathbb{R}^K$, with

$$
\varphi(t) = \left\{ \begin{array}{ll} t\ln(t) & \text{if } t > 0, \\ 0 & \text{if } t = 0, \\ +\infty & \text{otherwise.} \end{array} \right. \quad (7)
$$

We can now appeal to the concept of Fenchel-Legendre conjugate function. For a given convex function $f$, its conjugate function, denoted by $f^*$, is defined as $f^*(\boldsymbol{V}) = \sup_{\boldsymbol{U}}(\langle \boldsymbol{V}, \boldsymbol{U}\rangle - f(\boldsymbol{U}))$, where $\boldsymbol{V}$ is a dual variable [34, Def. 13.1] and $\langle \cdot, \cdot \rangle$ denotes the Euclidean scalar product on $\mathbb{R}^{K\times N}$. Since function $H$ is a proper, lower semicontinuous, convex function, according to the Fenchel-Moreau theorem [34, Thm. 13.37], the equality $H = (H^*)^*$ holds. Hence, it follows that

$$
-\lambda H(\boldsymbol{A}\boldsymbol{U}) = \lambda \inf_{\boldsymbol{V}} \{H^*(\boldsymbol{V}) - \langle \boldsymbol{V}, \boldsymbol{A}\boldsymbol{U}\rangle\}. \quad (8)
$$

Plugging (8) into (7), and using the expression of $H^*$ [35, Ex. 3.21], one obtains the following minimization problem with respect to the primal variable $\boldsymbol{U}$, the weights $\boldsymbol{W}$ and the dual variable $\boldsymbol{V}$:

$$
\underset{\boldsymbol{U},\boldsymbol{W},\boldsymbol{V}}{\text{minimize}} \quad \frac{1}{2}\sum_{k=1}^{K}\sum_{n=1}^{N} u_{n,k}\|\boldsymbol{w}_k - \boldsymbol{z}_n\|^2 + \lambda\sum_{k=1}^{K} e^{v_k-1} - \lambda\langle \boldsymbol{V}, \boldsymbol{A}\boldsymbol{U}\rangle, \quad \text{s.t} \quad (\mathrm{C}) \quad (9)
$$

**Handling the simplex constraint**   To deal efficiently with the simplex constraint in (9), we add an entropic barrier on soft assignment variables $\boldsymbol{u}_n$, leading to the following modified problem:

$$\operatorname*{minimize}_{\boldsymbol{U},\boldsymbol{W},\boldsymbol{V}} \quad \frac{1}{2}\sum_{k=1}^{K}\sum_{n=1}^{N}u_{n,k}\|\boldsymbol{w}_k-\boldsymbol{z}_n\|^2 + \lambda\sum_{k=1}^{K}e^{v_k-1} - \lambda\langle\boldsymbol{V},\boldsymbol{AU}\rangle + \underbrace{\sum_{n=1}^{N}\sum_{k=1}^{K}\varphi(u_{n,k})}_{\text{entropic barrier}}, \quad (10)$$

s.t       (C).

The last term in (10) acts as a barrier for imposing constraints $\boldsymbol{u}_n \geq 0$ and, at each iteration, yields closed-form updates of both the dual variables for constraints $\sum_{k=1}^{K} u_{n,k} = 1$ and assignments $\boldsymbol{U}$. This simplifies the iterative constrained-minimization steps over $\boldsymbol{U}$ in our algorithm below through simple closed-form softmax operations (derivation details provided in supplemental material).

**Block coordinate descent algorithm**   To minimize the cost function in (10), we pursue a block coordinate descent approach [36, 37], which is guaranteed to converge; see Proposition 1 below. At each iteration, we successively minimize the objective with respect to the variables $\boldsymbol{U}$, $\boldsymbol{V}$, and $\boldsymbol{W}$ respectively. To do so, we use the adjoint operator of $\boldsymbol{A}$, which we denote $\boldsymbol{A}^*$. Our algorithm is detailed in Algorithm 1.

---

**Algorithm 1:** **P**rim**Al** **D**ual Minimum **D**escription **LE**ngth (**PADDLE**)

---

Initialize $\boldsymbol{W}^{(0)}$ as the prototypes computed on the support, and $\boldsymbol{V}^{(0)} = \boldsymbol{0}$.
**for** $\ell = 1, 2, \ldots,$ **do**

$$\boldsymbol{U}^{(\ell)} = \operatorname{softmax}\left(-\frac{1}{2}\left(\|\boldsymbol{w}_k-\boldsymbol{z}_n\|^2\right)_{\substack{1\leq n\leq N\\1\leq k\leq K}} + \lambda\boldsymbol{A}^*\boldsymbol{V}^{(\ell-1)}\right),$$

$$v_k^{(\ell)} = 1 + \ln((\boldsymbol{AU}^{(\ell)})_k),\ \forall k \in \{1,\ldots,K\},$$

$$\boldsymbol{w}_k^{(\ell)} = \sum_{n=1}^{N}\boldsymbol{u}_{n,k}^{(\ell-1)}\boldsymbol{z}_n \Big/ \sum_{n=1}^{N}\boldsymbol{u}_{n,k}^{(\ell-1)},\ \forall k \in \{1,\ldots,K\}.$$

---

**Convergence guarantees**   In addition to its practical advantages in terms of implementation, our algorithm benefits from the convergence guarantee described in Proposition 1.

**Proposition 1.** *The sequence* $\left(\boldsymbol{U}^{(\ell)},\boldsymbol{W}^{(\ell)},\boldsymbol{V}^{(\ell)}\right)_{\ell\in\mathbb{N}^*}$ *generated by Algorithm 1 is bounded. Moreover, any of its cluster points is a critical point to the minimization problem in* (10).

A detailed proof of Proposition 1 is provided in the supplementary material.

## 5   Experiments

### 5.1   Experimental details

**Datasets.**   We deployed three datasets for few-shot classification: *mini*-Imagenet [38], *tiered*-Imagenet [18], and *i-Nat* [23]. A subset of the ILSVRC-12 data [38], *mini*-Imagenet is a standard few-shot benchmark, with $60,000$ color images of size $84 \times 84$ pixels [3]. It contains 100 classes, each represented with 600 images. We followed the standard split of 64 classes for base training, 16 for validation, and 20 for testing [39, 25]. The *tiered*-Imagenet is another standard few-shot benchmark, which is a larger subset of ILSVRC-12, with 608 classes and a total of $779,165$ color images of size $84 \times 84$ pixels. We used a standard split of 351 classes for base training, 97 for validation, and 160 for testing. Finally, the more realistic and challenging dataset *i-Nat* has 908 classes. We follow the split from [25, 4], with 227 ways at test-time.

**Task generation**   We build the few-shot tasks as follows. Let $s$ denotes the number of shots. We constitute the support set by sampling $s$ images according to the uniform distribution for each of the $K$ possible classes of the test set (i.e. $K = |\mathcal{Y}_{\text{test}}| = 20$, 160, and 227 for *mini*-ImageNet, *tiered*-ImageNet and *i-Nat*, respectively). For the query set, we first randomly pick $K_{\text{eff}} < K$ classes among the $K$ possible classes. We then randomly choose $|\mathbb{Q}|$ samples among the images that belong

to the $K_{\text{eff}}$ classes but do not appear in the support set. The right-hand side of Figure 1 depicts an example of a task, with the class color coding indicating whether a support class appears in the query set or not. The results presented below are obtained by fixing $|\mathbb{Q}| = 75$, as done in the literature. In the tables, we use a fixed $K_{\text{eff}} = 5$, but we also present the results over a larger range in Fig. 2. Given the difficulty of $K$-way tasks, with $K$ large, we consider 5-, 10- and 20-shot supervision to form the support set. Following standard evaluation, we report the accuracy, averaged across $10,000$ tasks.

**Hyper-parameters** We emphasize that PADDLE inference does not require any hyperparameter tuning since $\lambda$ in objective (10) is set to the number of samples in the query set $\mathbb{Q}$. This is typically not the case for other few-shot methods. For the tuning phase of those methods, we have followed the protocol of [10], which for each dataset, uses a 5-way, 5-shot scenario on the corresponding validation set. For *i-Nat*, because no validation split is provided, we reuse the hyper-parameters obtained from *tiered*-ImageNet.

**Feature extraction** To ensure the fairest comparison of methods, we use the pretrained checkpoints provided by the authors of [11]. The models are trained on $\mathcal{D}_{\text{base}}$ via a standard cross-entropy minimization with label smoothing. The label smoothing parameter is set to $0.1$, for 90 epochs, using a learning rate initialized to $0.1$ and divided by 10 at epochs $45$ and $66$. We use batch sizes of 256 for ResNet-18 and of 128 for WRN28-10. The images are resized to $84 \times 84$ pixels, both at training and evaluation time. Color jittering, random croping, and random horizontal flipping augmentations are applied during training.

| Method | Backbone | *mini*-ImageNet ($K$ =20) | | | *tiered*-ImageNet ($K$ =160) | | |
|---|---|---|---|---|---|---|---|
| | | 5-shot | 10-shot | 20-shot | 5-shot | 10-shot | 20-shot |
| Baseline [24] | ResNet-18 | 54.1 | 60.7 | 65.8 | 29.1 | 35.7 | 39.5 |
| LR+ICI [15] | | 54.7 | 62.0 | 67.2 | - | - | - |
| BD-CSPN [8] | | 49.6 | 54.1 | 55.6 | 24.0 | 26.7 | 26.0 |
| PT-MAP [7] | | 25.8 | 27.4 | 29.0 | 4.3 | 5.1 | 5.9 |
| LaplacianShot [4] | | 60.4 | 65.2 | 68.4 | 34.7 | 36.8 | 39.1 |
| TIM [11] | | **66.4** | 69.4 | 70.9 | 26.6 | 26.7 | 25.5 |
| $\alpha$-TIM [10] | | 63.5 | 67.4 | 71.7 | 38.7 | 44.2 | 48.4 |
| PADDLE (ours) | | 63.2 | **73.3** | **80.0** | **45.8** | **62.5** | **72.0** |
| Baseline [24] | WRN28-10 | 58.0 | 64.8 | 69.5 | 31.8 | 37.0 | 42.1 |
| LR+ICI [15] | | 57.2 | 64.3 | 70.9 | - | - | - |
| BD-CSPN [8] | | 51.2 | 55.5 | 58.4 | 23.7 | 24.9 | 23.8 |
| PT-MAP [7] | | 26.3 | 27.9 | 29.4 | 4.4 | 5.0 | 5.7 |
| LaplacianShot [4] | | 64.9 | 65.3 | 70.9 | 28.1 | 38.0 | 45.2 |
| TIM [11] | | **71.9** | **75.2** | 76.1 | 34.1 | 34.1 | 34.5 |
| $\alpha$-TIM [10] | | 68.3 | 72.5 | 75.8 | 41.9 | 46.1 | 51.8 |
| PADDLE (ours) | | 62.5 | 72.8 | **79.3** | **46.4** | **60.3** | **71.1** |

Table 1: Comparisons of state-of-the-art methods on *mini*-Imagenet and *tiered*-Imagenet, using the tasks generation process described in Sec. 2 with $K_{\text{eff}} = 5$. The metric is accuracy (in percentage). Results are averaged across 10,000 tasks. Results marked with '-' were intractable to obtain.

## 5.2 Results

**Main results** We compare the performances of our method with state-of-the-art few-shot methods. Our first experimental setting consists in fixing the number of effective classes $K_{\text{eff}}$ to 5. In Table 1, we evaluate the accuracy on *mini* and *tiered*-ImageNet for 5, 10, and 20 shots, while results on *i-Nat* are displayed in Table 2. Note that the *i-Nat* dataset comes with a unique support set for all tasks, with a varying number of shots (labeled samples) per class. Therefore, for *i-Nat*, we present the results separately. In the second experiment displayed in Figure 2 and 3, we plot the accuracy as a function of $K_{\text{eff}}$ on *mini*, *tiered*-ImageNet, and *i-Nat* for 5, 10, and 20 shots. Both experiments show that PADDLE is very competitive with the state-of-the-art methods on the proposed new practical few-shot setting. More importantly, the gap between PADDLE and the second best algorithm

increases significantly with (i) the number of shots and (ii) the number of possible classes $K$, when going from the dataset *mini*, which has a small number of test classes ($K = 20$), to *tiered* ($K = 160$) and finally to *i-Nat* ($K = 227$). From the first point, one may conclude that the proposed MDL formulation benefits better from additional support supervision. We hypothesize to this is due to its versatility, as it does not encode strong assumptions on the label statistics of the query set. Also, the second point could be explained along the same line. For instance, methods relying on a class-balance assumptions, such as TIM-GD or PT-MAP, observe a drastic drop in accuracy as $K$ increases, even falling below the inductive Baseline. This is expected because large values of $K$ corresponds to highly imbalanced classification problems for the query sets. For example, $K_{\mathrm{eff}} = 5$ effective classes on *tiered* corresponds to a relatively more drastic form of class imbalance than on *mini*, as it would mean only $5/160$ classes are represented in $\mathbb{Q}$, versus $5/20$. In contrast, PADDLE can deal with strongly imbalanced situations. Note that BD-CSPN [8] is the only baseline whose performance increases with $K_{\mathrm{eff}}$. This method is also among the most affected when the class overlap between the query and support sets decreases, i.e., when $K_{\mathrm{eff}}$ gets smaller (left side of the plots in Figure 2) and/or $K$ gets bigger (e.g. tiered-ImageNet, with $K = 160$). This behaviour might be due to the fact that BD-CSPN encodes a strong prior, assuming the support and query classes match perfectly.

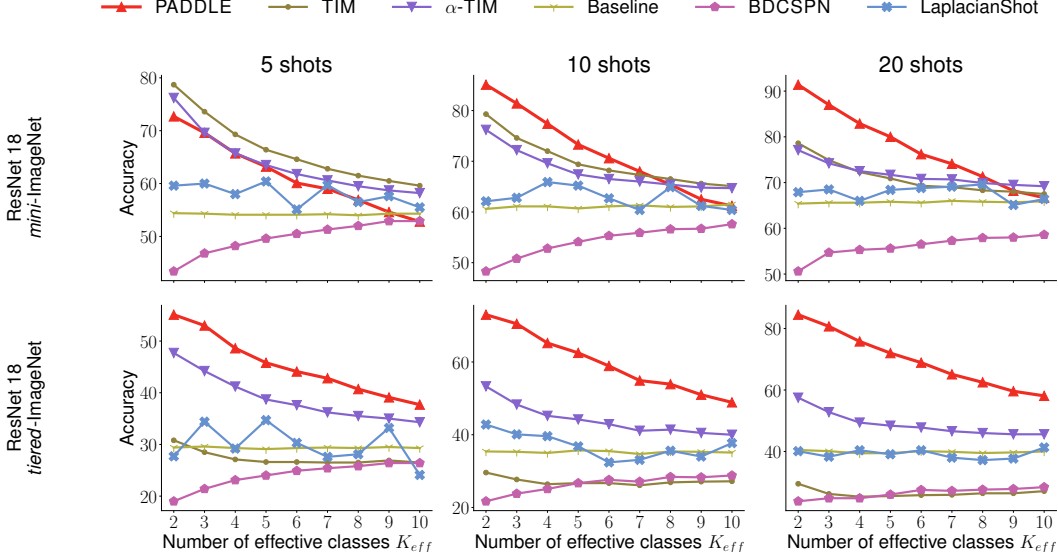

Figure 2: Evolution of the accuracy as a function of $K_{\mathrm{eff}}$. Each row represents a dataset, and each column a fixed number of shots. All methods use the same ResNet-18 network. Results are averaged across 10,000 tasks.

| Method | *i-Nat* ($K = 227$) |
|---|---|
| Baseline [24] | 58.2 |
| BD-CSPN [8] | 57.6 |
| PT-MAP [7] | 6.8 |
| LaplacianShot [4] | 43.3 |
| TIM [11] | 37.5 |
| $\alpha$-TIM [10] | 66.7 |
| PADDLE (ours) | **84.3** |

Table 2: Similarly to Table 1, results are provided with a fixed $K_{\mathrm{eff}} = 5$ on *i-Nat* with 227-ways tasks.

Figure 3: Similarly to Fig. 2, we plot the performances of the methods as functions of $K_{\mathrm{eff}}$ on *i-Nat* with 227-ways tasks.

**Ablation on the objective** We hereby ablate on the importance of the *partition-complexity* term in Eq. (2). Removing this high-order term yields a partially-supervised version of $K$-means, which can be optimized effortlessly through iterative closed-form alternating steps. We provide the comparison

between this approach (*without* the partition-complexity term) and PADDLE (*with* the partition complexity term) in Table 3. On can observe that PADDLE systematically outperforms its $K$-Means counterpart, with absolute differences in accuracy reaching up to 30% + on challenging scenarios.

| Partition complexity term in Problem 2 | *mini* ($K$ =20) | | | *tiered* ($K$ =160) | | | **i-Nat** ($K$ =227) |
|---|---|---|---|---|---|---|---|
| | 5-shot | 10-shot | 20-shot | 5-shot | 10-shot | 20-shot | |
| Without | 49.6 | 57.2 | 63.6 | 28.3 | 35.8 | 39.5 | 57.5 |
| With | **63.2** | **73.3** | **80.0** | **45.8** | **62.5** | **72.0** | **84.3** |

Table 3: **Importance of the partition-complexity term.** Results are computed using a ResNet-18 and $K_{\text{eff}}$ =5, and averaged across 10,000 tasks. Without the *partition-complexity* term in Eq. (2), the algorithm has little incentive to be parsimonious in its choice of the effective classes. As a matter of fact, removing this term reduces Problem (2) to a partially-supervised K-means algorithm, notoriously known to encourage balanced solutions across the classes.

**Ablation on the optimization procedure**   As a second ablation, we propose to compare our proposed Alternating Minimization Algorithm **??** to a straightforward first-order approach. More precisely, as a comparison, we directly optimize Eq. (2) through Projected Gradient Descent, denoted as PGD , where a simplex projection step [40] takes place after every iteration to ensure that the simplex constraints on $U$ remains satisfied. For PGD , we use Adam [41] with $\alpha = 0.001$. We found this to be the highest learning rate that leads to convergence. We also add the convergence time of the second best competing method, $\alpha$-TIM. Results are provided in Fig. 4. PGD is much slower than PADDLE , with a ratio between run times neighboring an order of magnitude. Additionally, as hinted from the last points of the PGD curve, oscillations seem to occur, indicating that a more sophisticated learning rate policy would be necessary to achieve a clear convergence.

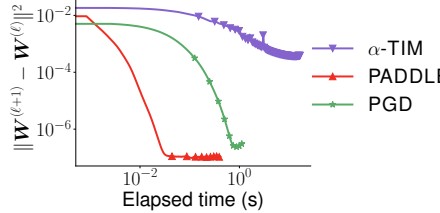

| | Time to convergence (s) | Accuracy |
|---|---|---|
| $\alpha$-TIM | N/A | 61.0 |
| PGD | $6.1 \times 10^{-1}$ | 53.0 |
| PADDLE | $2.6 \times 10^{-2}$ | 77.0 |

Figure 4: **Comparison of PADDLE and PGD in terms of convergence speed** (Left) The chosen criterion as a function of the elapsed time on a randomly chosen 20-shot, $K_{\text{eff}}$ =5 task sampled from *tiered*-ImageNet. Both methods are run on the same machine, with markers displayed every 100 iterations. (Right) time to convergence and accuracy, after reaching a value of $10^{-6}$ on the criterion.

## Conclusion and limitations

We presented a practical few-shot setting where the number of candidate classes can be much larger than the number of classes that appear effectively in the query set. Our setting is an instance of highly imbalanced classification, with large numbers of ways and potentially irrelevant supervision from the support set. We observed much higher gaps across transductive few-shot methods, some of which fall below the simple inductive baseline. As a solution, we introduced PADDLE , which casts this challenge as a partially-supervised MDL partitioning problem, interpreting the number of unique classes found as a measure of model complexity. PADDLE is hyperparameter-free, and remains competitive over various state-of-the-art methods, settings and datasets, without the need for any tuning. However, we do not advocate our method as the one-fits-all, ultimate solution to the extremely challenging few-shot problem. For instance, PADDLE would not be the best performing method in situations where $K_{\text{eff}}$ would approach $K$. In applications where one has knowledge about the label statistics of the query set (e.g. class balance), other methods encoding such a knowledge could be more appropriate. More generally, our setting can be seen as a particular case of class-distribution shift between the support and query sets. Interesting future works could complement our current study by overlaying other forms of shifts, e.g. feature shifts [42, 43], to our setting.

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
