# Supplementary material

## A  Closed-form updates of the assignment variables

In this section, we provide more details on the derivation of the closed-form update of variable $\boldsymbol{U}$ at each iteration. Let $F$ be the defined as the cost function in (10) and let $\partial F_{\boldsymbol{u}_n}(\boldsymbol{U}, \boldsymbol{W}, \boldsymbol{V})$ denote the Moreau subdifferential of $F$ at $(\boldsymbol{U}, \boldsymbol{W}, \boldsymbol{V})$ with respect to variable $\boldsymbol{u}_n$. We define $\psi$ as

$$(\forall \boldsymbol{x} = (x_k)_{1 \le k \le K} \in \mathbb{R}^K) \quad \psi(\boldsymbol{x}) = \begin{cases} \displaystyle\sum_{k=1}^{K} x_k \ln(x_k) - \frac{x_k^2}{2} & \text{if } \boldsymbol{x} \in \Delta_K, \\ +\infty & \text{otherwise.} \end{cases} \tag{11}$$

It is well known that the proximity operator of $\psi$ (see [34, Chap. 24] for a definition) is the softmax operator [44, Ex. 2.23].

At each step of the algorithm, $\boldsymbol{u}_n$ is updated according to:

$$0 \in \partial F_{\boldsymbol{u}_n}(\boldsymbol{U}, \boldsymbol{W}, \boldsymbol{V})$$

$$\iff \quad 0 \in \frac{1}{2}\left(\|\boldsymbol{w}_k - \boldsymbol{z}_n\|^2\right)_{1 \le k \le K} - \lambda[\boldsymbol{A}^*\boldsymbol{V}]_n + \boldsymbol{u}_n + \partial_\psi(\boldsymbol{u}_n),$$

$$\iff \quad -\frac{1}{2}\left(\|\boldsymbol{w}_k - \boldsymbol{z}_n\|^2\right)_{1 \le k \le K} + \lambda[\boldsymbol{A}^*\boldsymbol{V}]_n - \boldsymbol{u}_n \in \partial_\psi(\boldsymbol{u}_n),$$

$$\iff \quad \boldsymbol{u}_n = \text{softmax}\left(-\frac{1}{2}\left(\|\boldsymbol{w}_k - \boldsymbol{z}_n\|^2\right)_{1 \le k \le K} + \lambda[\boldsymbol{A}^*\boldsymbol{V}]_n\right), \tag{12}$$

where we used the definition of the proximity operator [34, Eq. 24.2] to obtain (12). We thus retrieve the update in Algorithm 1.

## B  Proof of Proposition 1

Our proof relies on the convergence result established in [45]. Given a convex set $X$, we denote $\iota_X$ the indicator function of $X$, i.e. $\iota_X(x) = 0$ if $x \in X$, $\iota_X(x) = +\infty$ otherwise. We rewrite problem 10 as the minimization of the following cost:

$$F(\boldsymbol{U}, \boldsymbol{W}, \boldsymbol{V}) = \frac{1}{2}\sum_{k=1}^{K}\sum_{n=1}^{N} u_{n,k}\|\boldsymbol{w}_k - \boldsymbol{z}_n\|^2 + \lambda\sum_{k=1}^{K} e^{v_k - 1} - \lambda\langle\boldsymbol{V}, (\boldsymbol{A}\boldsymbol{U} + \epsilon\mathbf{1}_K)\rangle$$
$$+ \sum_{n=1}^{N}\sum_{k=1}^{K}\varphi(u_{n,k}) + \iota_C(\boldsymbol{U}), \tag{13}$$

where we have introduced an additional parameter $\epsilon > 0$, the role of which will become clearer in the rest of the proof. The optimum of the cost function $F(\boldsymbol{U}, \boldsymbol{W}, \cdot)$ for given $\boldsymbol{U} \in C$ and $\boldsymbol{W} \in (\mathbb{R}^d)^K$ is reached when

$$\boldsymbol{V} = \mathbf{1}_K + \ln(\boldsymbol{A}\boldsymbol{U} + \epsilon\mathbf{1}_K) \in \mathbb{V}_\epsilon = [1 + \ln\epsilon, 1 + \ln(1 + \epsilon)]^K. \tag{14}$$

Thus, minimizing $F$ is actually equivalent to minimizing

$$\tilde{F}(\boldsymbol{U}, \boldsymbol{W}, \boldsymbol{V}) = \frac{1}{2}\sum_{k=1}^{K}\sum_{n=1}^{N} u_{n,k}\|\boldsymbol{w}_k - \boldsymbol{z}_n\|^2 + \lambda\sum_{k=1}^{K} e^{v_k - 1} - \lambda\langle\boldsymbol{V}, (\boldsymbol{A}\boldsymbol{U} + \epsilon\mathbf{1}_K)\rangle$$
$$+ \sum_{n=1}^{N}\sum_{k=1}^{K}\varphi(u_{n,k}) + \iota_C(\boldsymbol{U}) + \iota_{\mathbb{V}_\epsilon}(\boldsymbol{V}). \tag{15}$$

The following algorithm for minimizing $\tilde{F}$ turns out to be a simple modified version of PADDLE (see Algorithm 1):

---
**Algorithm 2:** Alternating algorithm for minimizing $\tilde{F}$

---
Initialize $\boldsymbol{W}^{(0)}$ as the prototypes computed on the support, and $\boldsymbol{V}^{(0)} = \boldsymbol{0}$.
**for** $\ell = 1, 2, \dots,$ **do**

$$\boldsymbol{U}^{(\ell)} = \text{softmax}\left(-\frac{1}{2}\left(\|\boldsymbol{w}_k - \boldsymbol{z}_n\|^2\right)_{\substack{1 \le n \le N \\ 1 \le k \le K}} + \lambda \boldsymbol{A}^* \boldsymbol{V}^{(\ell-1)}\right),$$

$$v_k^{(\ell)} = 1 + \ln((\boldsymbol{A}\boldsymbol{U}^{(\ell)})_k + \epsilon), \ \forall k \in \{1, \dots, K\},$$

$$\boldsymbol{w}_k^{(\ell)} = \sum_{n=1}^{N} \boldsymbol{u}_{n,k}^{(\ell-1)} \boldsymbol{z}_n \ \Big/ \ \sum_{n=1}^{N} \boldsymbol{u}_{n,k}^{(\ell-1)}, \ \forall k \in \{1, \dots, K\}.$$

---

According to [45, Thm 4.1], if the following assumptions are satisfied:

1. The set $\left\{ (\boldsymbol{U}, \boldsymbol{W}, \boldsymbol{V}) \ : \ \tilde{F}(\boldsymbol{U}, \boldsymbol{W}, \boldsymbol{V}) \le \tilde{F}(\boldsymbol{U}^{(0)}, \boldsymbol{W}^{(0)}, \boldsymbol{V}^{(0)}) \right\}$ is compact;

2. $\tilde{F}$ is continuous on $C \times (\mathbb{R}^d)^K \times \mathbb{V}_\epsilon$;

3. At each iteration $\ell$, the partial functions $\tilde{F}(\cdot, \boldsymbol{W}^{(\ell)}, \boldsymbol{V}^{(\ell)})$, $\tilde{F}(\boldsymbol{U}^{(\ell+1)}, \cdot, \boldsymbol{V}^{(\ell)})$ and $\tilde{F}(\boldsymbol{U}^{(\ell+1)}, \boldsymbol{W}^{(\ell+1)}, \cdot)$ admit a unique minimizer,

then the sequence generated by the algorithm is bounded and every of its cluster points is a coordinatewise minimizer of $\tilde{F}$. We now show that the above assumptions hold.

1. Let us show that $\tilde{F}$ is coercive. We derive a lower bound on $\tilde{F}$ using the Cauchy-Schwarz inequality:

$$\tilde{F}(\boldsymbol{U}, \boldsymbol{W}, \boldsymbol{V}) \ge \frac{1}{2} \sum_{k=1}^{K} \sum_{n=|\mathbb{Q}|+1}^{N} y_{n,k} \|\boldsymbol{w}_k - \boldsymbol{z}_n\|^2 + \lambda \sum_{k=1}^{K} e^{v_k - 1} - \lambda \|\boldsymbol{V}\| \|\boldsymbol{A}\boldsymbol{U}\|$$

$$- \epsilon \langle \boldsymbol{V}, \boldsymbol{1}_K \rangle + \sum_{n=1}^{N} \sum_{k=1}^{K} \varphi(u_{n,k}) + \iota_C(\boldsymbol{U}) + \iota_{\mathbb{V}_\epsilon}(\boldsymbol{V}). \quad (16)$$

Since the functions $\boldsymbol{U} \mapsto \|\boldsymbol{A}\boldsymbol{U}\|$ and $\boldsymbol{U} \mapsto \sum_{n=1}^{N} \sum_{k=1}^{K} \varphi(u_{n,k})$ are continuous on the compact set $C$, there exist constants $\mu$ and $\theta$ such that

$$\tilde{F}(\boldsymbol{U}, \boldsymbol{W}, \boldsymbol{V}) \ge \frac{1}{2} \sum_{k=1}^{K} \sum_{n=|\mathbb{Q}|+1}^{N} y_{n,k} \|\boldsymbol{w}_k - \boldsymbol{z}_n\|^2 + \lambda \sum_{k=1}^{K} e^{v_k - 1} - \theta \|\boldsymbol{V}\|$$

$$- \epsilon \langle \boldsymbol{V}, \boldsymbol{1}_K \rangle + \mu + \iota_C(\boldsymbol{U}) + \iota_{\mathbb{V}_\epsilon}(\boldsymbol{V}). \quad (17)$$

The lower bound obtained in (17) is separable in $(\boldsymbol{U}, \boldsymbol{W}, \boldsymbol{V})$. The term with respect to variable $\boldsymbol{W}$ is coercive when, for every $k \in \{1, \dots, K\}$, there exists $n \in \{|\mathbb{Q}| + 1, \dots, N\}$ such that $y_{n,k} > 0$. In other words, it is coercive if the support set includes at least one example of each class, which is a reasonable assumption. The terms with respect to variables $\boldsymbol{U}$ and $\boldsymbol{V}$ are clearly coercive too. Hence, the cost function $\tilde{F}$ is coercive. Finally, since $\tilde{F}$ is lower semi-continuous, condition 1. is satisfied.

2. The continuity of $\tilde{F}$ on $C \times \mathbb{R}^{k \times d} \times \mathbb{V}_\epsilon$ is clear.

3. Let $\ell \in N^*$. We already proved in Appendix A that the partial function with respect to variable $\boldsymbol{U}$ has a unique minimizer. It follows from the same arguments as above that the partial function with respect to $\boldsymbol{W}$ is strictly convex, continuous, and coercive as soon as the support set contains at least one example of each class. Hence, it admits a unique minimizer. Regarding the partial function with respect to variable $\boldsymbol{V}$, we first remark that given the definition of the softmax operator, $\boldsymbol{A}\boldsymbol{U}^{(\ell+1)}$ is necessarily strictly positive component-wise. Up to some additive term independent of $\boldsymbol{V}$, the partial function reads

$$\boldsymbol{V} \mapsto \lambda \sum_{k=1}^{K} \left( e^{v_k - 1} - v_k([\boldsymbol{A}\boldsymbol{U}^{(\ell+1)}]_k + \epsilon) + \iota_{[\ln \epsilon, \ln(1+\epsilon)]}(v_k - 1) \right). \quad (18)$$

The latter function is strictly convex, lower-semicontinuous, and coercive, which concludes the proof.

Note that, since

$$v_k \mapsto \lambda \left( e^{v_k - 1} - v_k([\boldsymbol{A}\boldsymbol{U}^{(\ell+1)}]_k + \epsilon) \right) \tag{19}$$

is decreasing on $]-\infty, 1 + \ln([\boldsymbol{A}\boldsymbol{U}^{(\ell+1)}]_k + \epsilon)]$ and increasing on $[1 + \ln([\boldsymbol{A}\boldsymbol{U}^{(\ell+1)}]_k + \epsilon), +\infty[$, the resulting cluster points are also coordinatewise minimizers of $F$.

In summary, PADDLE can be understood as the limit case of Algorithm 2 when $\epsilon$ goes to zero. This simplification is justified by the fact that $\epsilon$ can be chosen arbitrarily small and that we did not observe any change in practical behaviour of the proposed algorithm by setting $\epsilon = 0$.

## C  Label cost relaxation

The plot in Figure 5 illustrates in the case $K = 2$ how our model-complexity term in (2) could be viewed as a continuous relaxation of the discrete label cost function defined in (3).

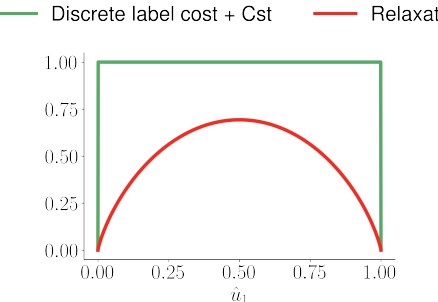

Figure 5: Label cost as a function of $\hat{u}_1$ and our proposed relaxation $\hat{u}_1 \mapsto -\hat{u}_1 \ln(\hat{u}_1) - (1 - \hat{u}_1) \ln(1 - \hat{u}_1)$.

## D  Plots obtained using WRN backbone

In Figure 6, we provide additional comparisons of PADDLE with state-of-the-art methods using a WRN28-10 network. We report the accuracy obtained for each method as a function of $K_{\text{eff}}$. These plots point to the same conclusions drawn in Section 5.

## E  About the hyper-parameter in our method

As discussed in Section 3, PADDLE does not require parameter tuning. In Figure 7, we investigate the optimal value of parameter $\lambda$ in (10) as a function of the size of the query set, for 3 different values of $K_{\text{eff}}$. We observe that the optimal value of $\lambda$ increases linearly with $|\mathbb{Q}|$. As it could be expected, the higher the level of class imbalance ($K_{\text{eff}} = 2$), the higher the optimal value of $\lambda$ (w.r.t. its theoretical value). On the contrary, when the query is better balanced ($K_{\text{eff}} = 10$), the optimal value of $\lambda$ is slightly under its theoretical value. However, Figure 8 shows that the gap of performance when using the theoretical value of $\lambda$ instead of the optimal one, is only of the order of a few percents.

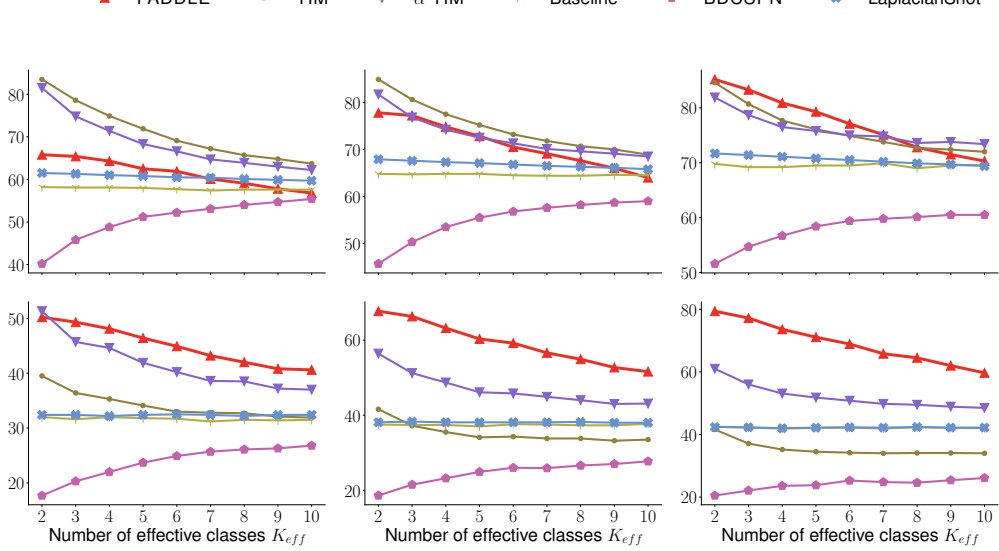

Figure 6: Evolution of the accuracy as a function of $K_{\text{eff}}$. Each row represents a dataset, and each column a fixed number of shots. All methods use the same WRN28-10 network. Results are averaged across 10,000 tasks.

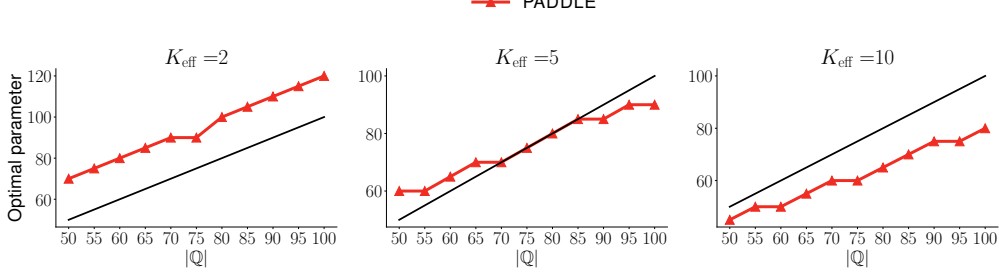

Figure 7: Evolution of the optimal parameter $\lambda$ (i.e. the one with which the best accuracy is reached) as a function of $|\mathbb{Q}|$. Each column represents a fixed number of effective classes. The black line represents the identity function. The results were computed on the *tiered* dataset with a Resnet18 as a backbone.

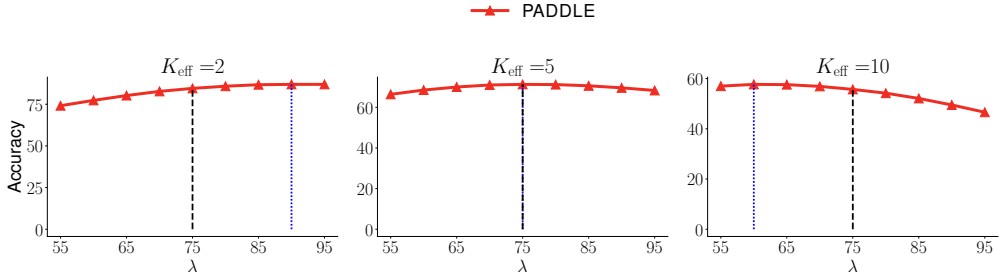

Figure 8: Evolution of the accuracy as a function of $\lambda$. Each column represents a fixed number of effective classes. The results were computed on the *tiered* dataset with a Resnet18 as a backbone, and the size query set was fixed to $|\mathbb{Q}| = 75$. The blue dotted line represents the optimal value of $\lambda$ while the black dashed line represents the theoritical value of $\lambda$, i.e. $\lambda = |\mathbb{Q}|$.