# OpenReview forum: "Towards Practical Few-shot Query Sets: Transductive Minimum Description Length Inference"
_NeurIPS.cc/2022/Conference — NeurIPS 2022 Accept_

### Official Review · Reviewer_m7wN · 2022-07-11

**Rating:** 5
**Confidence:** 3
**Soundness:** 3 good
**Presentation:** 2 fair
**Contribution:** 3 good

**Summary:**

This paper considers a more practice problem about the classes between the support and query sets, i.e., imbalanced K-way classification. A PADDLE method is proposed to solve the problem, which balances data-fitting accuracy and model complexity for a given few-shot task. Experiments show that the method achieves competitive performances in the standard few-shot datasets.

**Questions:**

1. Important related work is missing from this paper, for example: transductive few-shot learning and few-shot image classification.
2. Some transductive few-shot learning methods should be compared in table 1, such as FEAT [1], SIB [2], DPGN [3].
3. The equations (2) and (C) do not match, u_{n} does not appear in the (2). Is the class prototypes W a learned parameter? if not, why minimize the W?
4. The optimization objective of transductive few-shot learning methods has a cross-entropy loss and entropy-based loss, but equation (2) has data-fitting accuracy and partition complexity.  What is the relationship between them？


[1] Few-Shot Learning via Embedding Adaptation with Set-to-Set Functions

[2] Empirical Bayes Transductive Meta-Learning with Synthetic Gradients

[3] DPGN: Distribution Propagation Graph Network for Few-shot Learning

**Limitations:**

Yes

**Strengths And Weaknesses:**

Strengths:

This paper considers a very interesting problem, i.e., the query-set classes are unknown and do not match exactly the support-set classes, but just belong to a much larger set of possible classes.

Weaknesses

The clarity for this paper should be improved and related work should be discussed.

---

> ### Author Response · Authors · 2022-08-02
> **Answer to reviewer #m7wN (Part 1)**
>
> **Q:** Some transductive few-shot methods should be evaluated and added to Table 1, such as FEAT [Ye et al., CVPR’20], SIB [Yu et al., ICLR’20], DPGN [Yang et al., CVPR’20].
>
> **A:** Thanks for pointing to these additional baselines! We added evaluations of FEAT [Ye et al., CVPR’20], SIB [Yu et al., ICLR’20], and DPGN [Yang et al., CVPR’20] in our new setting (using the official GitHub repositories of these methods). The results of these new evaluations on Mini-Imagenet are summarized in the table below (we will update the paper with these):
>
> | **Method** | **5-shot** | **10-shot** | **20-shot** |
> | :-------------------- | :-------------------- | :-------------------- | :-------------------- |
> | **FEAT** | 55.8 (&#8595;26.3) | 62.1 (&#8595;21.4) | 66.2 (&#8595;19.5) |
> | **SIB** | 40.1 (&#8595;40.4) | 41.4 (&#8595;40.5) | 42.1 (&#8595;40.6) |
> | **DPGN** | 26.3 (&#8595;58.3) | N.E. | N.E. |
> | **PADDLE** | **63.2** (&#8595;13.1) | **73.3** (&#8595;6.0) | **80.0** (&#8595;2.0) |
>
> N.E.: *Not Evaluable* because of memory issues during the training (as in our setting the number of ways is larger than the value typically used).
>
> The arrows &#8595; indicate the drop in accuracy of the methods evaluated on our realistic few-shot setting compared to the standard setting with balanced 5-way tasks.
>
> Please note that, for SIB and DPGN, we had to re-train the networks because the provided pre-trained models were only adapted to a fixed number of ways. Consistently with the observations made in Table 1 for transductive few-shot methods, the performances of FEAT, SIB, and DPGN also drop in our new setting and are, overall, below the state-of-the-art methods we evaluated in the paper. Indeed, we tried to prioritize evaluations of the best-performing methods, but we agree that adding more baselines could only strengthen the experiments.
>
> ---------
> **Q:** The equations (2) and (C) do not match, $\boldsymbol{u}_{n}$ does not appear in the (2).
>
> **A:** In fact, $\\boldsymbol{u}\_{n} = (u\_{n,k})\_{1\leq k \leq K} \in \\{0,1\\}^K$ which we defined in line 119 (beginning of the formulation in Section 3, where we introduced our notations). Therefore, simplex vectors $\\boldsymbol{u}\_{n}$ in Eq. (C) are just concatenation of assignment variables $u\_{n,k}$ in Eq. (2). This enables a more compact notation of the simplex constraints in (C), avoiding summations and inequalities. For better readability, we will move notation $\\boldsymbol{u}\_{n} = (u\_{n,k})\_{1\leq k \leq K} \in \\{0,1\\}^K$ just before Eqs. (2) and (C).
>
> -------
>
> **Q:** Is the class prototypes $\\boldsymbol{W}$ a learned parameter? if not, why minimize the W?
>
> **A:** Yes, indeed, class prototypes $\\boldsymbol{W}$ are learned during inference. We alternate minimizations of the objective in Eq. (2) w.r.t both prototypes $\\boldsymbol{W}$ (iterative updates $\\mathbf{w}\_k^{(\\ell)}$ in the Algorithm detailed in line 222) and assignment variables $\\boldsymbol{U}$.
>
> --------
>
> **Q:**  Important related work is missing, for example, transductive few-shot learning.
>
> **A:** We agree with Reviewer #m7wN that providing more details on the objectives of transductive few-shot methods will be more informative. Reviewer #7GF also pointed to this. Please note that, in the second paragraph of the introduction, we cited over 15 transductive few-shot methods, and categorized them into different types of approaches: Graph regularization (e.g. [Ziko et al., ICML'20]), optimal transport (e.g. [Lazarou et al., ICCV'21]), feature transformations (e.g. [Liu et al., ECCV'20]), information maximization (e.g [Veilleux et al., NeurIPS'21]), and transductive batch normalization (e.g [Bronskill et al., ICML'20]), among other works. We concede, however, that a full-fledged related-work section, in which we move this introduction discussion and further augment it with more details the objectives of transductive few-shot methods, will be good. We will add to the paper the paragraph "Additional discussions on transductive few-shot objectives", which can be found in our answer to reviewer #7GFu.

---

> > ### Author Response · Authors · 2022-08-02
> > **Answer to reviewer #m7wN (Part 2)**
> >
> >
> > **Q:** The optimization objective of transductive few-shot learning methods has a cross-entropy loss and entropy-based loss, but equation (2) has data-fitting accuracy and partition complexity. What is the relationship between them? }
> >
> > **A:** In fact, unlike the existing objectives that use cross-entropy, support supervision in our case comes into the forms of constraints (C), which avoids an additional hyper-parameter to weight the cross-entropy. Please notice that these constraints influence indirectly prototypes $\\boldsymbol{W}$ that are learned during optimization.
> >
> > The entropy used in transductive methods is based on sample-wise prediction, i.e., minimizing the entropy for each query sample encourages a confident prediction for the sample. In our case, the partition complexity term is different as it measures the entropy of the overall partition, and minimizing it penalizes the number of classes that effectively appear in the partition ($K\_{\text{eff}}~$). In Figure E in the anonymous link above, we provide a graphical illustration as to how our partition-complexity term could be viewed as an interesting continuous relaxation of the discrete label count (Eq. 3) used in MDL-based clustering methods. To the best of our knowledge, our work is the first to explore MDL principles in the context of transductive few-shot learning.

---

### Official Review · Reviewer_ig28 · 2022-07-12

**Rating:** 6
**Confidence:** 3
**Soundness:** 3 good
**Presentation:** 3 good
**Contribution:** 3 good

**Summary:**

This paper introduced a PrimAl Dual Minimum Description LEngth (PADDLE) formulation, which balances data-fitting accuracy and model complexity for a given few-shot task, under supervision constraints from the support set.It is hyper-parameter free, and could be applied on top of any base-class training. Furthermore, this paper derive a fast block coordinate descent algorithm for optimizing our objective, with convergence guarantee, and a linear computational complexity at each iteration. Comprehensive experiments show highly competitive performances of this method, more so when the numbers of possible classes in the tasks increases.

**Questions:**

1. Based on their paper, I find that support is not very useful for few-shot learning compared with query set. I wonder if the randomly generated support sets would bring about good performance.

**Ethics Review Area:**

["I don’t know"]

**Limitations:**

They have  adequately addressed the limitations and potential negative societal impact of their work.

**Strengths And Weaknesses:**

Strengths:

1.The formulation proposed by this method is hyper-parameter free, and could be applied on top of any base-class training. This is very rare and significant.

2.The writing of the article is concise and clear. The formula is rigorous and the logic is clear.

3.Experiments are sufficient and strongly confirm the validity of the proposed method.

4.The proposed few-shot setting is practical and will have a positive impact on many related work.

Weaknesses:

1. There seems to be a lack of a framework to give a visual overview of the proposed approach.

2. On mini-ImageNet dataset and set to 5-shot, the proposed method is slightly inferior to TIM [1]. What is the reason for this? Why didn't this happen on the tiered-ImageNet dataset?

3. Experimental Settings are not clearly explained, including software and hardware conditions, the settings of each hyper-parameter, etc.

Reference

[1] M. Boudiaf, I. M. Ziko, J. Rony, J. Dolz, P. Piantanida, and I. Ben Ayed, “Transductive information maximization for few-shot learning,” in Neural Information Processing Systems (NeurIPS), 2020.

---

> ### Author Response · Authors · 2022-08-02
> **Answer to reviewer #ig28**
>
> **Q:** Lack of a framework to give a visual overview of the proposed approach.
>
> **A:** Thank you for the suggestion! We agree that a general graphical illustration would be helpful to readers. We produced this in Fig. A at the anonymous link above (Top-level answer to all reviewers), and we will add it to the paper.
>
> ---------------
>
> **Q:** On mini-ImageNet dataset and set to 5-shot, the proposed method is slightly inferior to TIM [Boudiaf et al., NeurIPS'20]. What is the reason for this? Why didn't this happen on the tiered-ImageNet dataset?
>
> **A:** These differences between Mini-Imagenet and Tiered-Imagenet could be explained by the degree of class balance in the considered few-shot tasks. Specifically, the tasks for which the methods in Table 1 were evaluated are more balanced for Mini-Imagenet than for Tiered-Imagenet. Indeed, in the case of Mini-Imagenet, the query set consists of $K_{\rm eff}=5$ unknown classes among the $K=20$ possible, while for Tiered-Imagenet, we have $K_{\rm eff}=5$ and $K=160$. Since TIM encodes a prior on class balance, it performs very well on Mini-Imagenet. It undergoes, however, a drastic drop in accuracy on Tiered-Imagenet and i-Nat. Please note that the performances of TIM are briefly discussed in Section 5.2, at the end of the main results paragraph.
>
> --------------
>
> **Q:** More details on the experimental settings, including software/hardware conditions, hyper-parameters, etc.}
>
> **A:** We provide more details here:
>
> - Few-shot task generation: Please refer to the additional details we provided to reviewer #7GFu.
>
> - Software/hardware conditions: The code was developed under python 3.8.3 and PyTorch 1.4.0, and all the experiments were executed on a single Tesla V100-SXM2 GPU with 32GB of Memory.
>
> - Base training: We adopted a standard cross-entropy training, with an experimental setting that is commonly used in the few-shot literature. Please note that we briefly provided the base-training details in the feature-extraction paragraph, page 7.
>
> - Hyper-parameters: For PADDLE, the unique parameter $\lambda$ was set automatically to $|\mathbb{Q}|$ for all experiments, according to the theoretical insight we provided in Section 3. For other inference methods that required hyper-parameter tuning, we followed standard protocols in the few-shot literature (e.g., [Veilleux et al., NeurIPS'21], among others): The optimal hyper-parameters are found on a validation set, in which the classes are completely different from those observed in testing. Please note that details on the validation sets are provided in the Datasets paragraph, page 6, and more details on hyper-parameters tuning are provided in the Hyper-parameters paragraph, page 7.
>
> --------
>
> **Q:** Would randomly generated support sets bring good performance?
>
> **A:** In fact, the support sets are also randomly generated and the support images vary from one task to another, across the 10,000 tasks randomly generated for evaluation. For each task, we randomly sample $n$ images (shots) per each possible class, with $n$ substantially smaller than the total number of images available per class. The difficulty might change from one task to another due to variations in the support sets. We believe evaluation over 10,000 tasks (a standard choice in the few-shot literature) provides a good proxy for the overall performances of the methods.

---

### Official Review · Reviewer_7GFu · 2022-07-14

**Rating:** 7
**Confidence:** 3
**Soundness:** 4 excellent
**Presentation:** 3 good
**Contribution:** 3 good

**Summary:**

This paper proposes an algorithm called PADDLE to perform transductive few-shot classification where the classes in the query set may not entirely overlap with the classes in the support set. PADDLE consists of a partially supervised clustering optimization that balances partition complexity and the fit of prototypes to the data. An efficient block coordinate descent algorithm to solve this optimization is proposed and analyzed. Experiments on several few-shot classification benchmark datasets in the proposed transductive setting show that PADDLE often exhibits strong gains in accuracy relative to transductive baseline methods.

**Questions:**

- How does the current work differ from other transductive few-shot approaches such as those proposed by Ziko et al. (2020) and Veilleux et al. (2021)?
- Please describe the task generation procedure in more detail. For example, how were the classes in the query set sampled? Are they guaranteed to partially overlap with the support set in some way?
- In figure 1, is there any insight as to why the accuracy of some baseline methods actually increases with $K_{eff}$?
- Is there any empirical evidence that setting $\lambda = |\mathbb{Q}|$ works better than alternatives?

**Limitations:**

Limitations could be improved by discussing the following:
- How can $K$ be set in a real-world scenario where the number of novel classes at test time is unknown?
- When is the transductive setting likely to not hold? Is there any recourse when there exists only a single query example?

**Strengths And Weaknesses:**

Strengths
- The few-shot setting investigated in this work is timely and realistic due to removing the assumption of perfect class overlap between support and query sets.
- The proposed algorithm is highly effective in the proposed transductive scenario. Performance is often quite significantly better than baselines.
- Experiments are thorough and informative. Sensible ablations demonstrate the utility of the partition complexity term and the proposed block coordinate descent algorithm.
- Analysis of the complexity of the algorithm and its convergence properties are provided.
- The writing is clear and insightful, drawing connections to classical algorithms.

Weaknesses
- It is unclear how the proposed method relates to previously proposed transductive few-shot algorithms and to existing MDL-based clustering algorithms. A related work section should be added.
- Experiments show the effectiveness of PADDLE in the proposed imbalanced transductive scenario but they do not show the limitations of PADDLE a more balanced setting.
- The task generation procedure is not described in sufficient detail.

---

> ### Author Response · Authors · 2022-08-02
> **Answer to reviewer #7GFu (Part 1)**
>
> **Q:** How does the current work differ from other transductive few-shot approaches such as those proposed by Ziko et al. (2020) and Veilleux et al. (2021)?
>
> **A:** We agree with Reviewer #7GFu that providing more details on the objectives of transductive few-shot methods will be more informative and will further emphasize the difference with our MDL-like formulation.
> Please note that, in the second paragraph of the introduction, we cited over 15 transductive few-shot methods, and categorized them into different types of approaches: Graph regularization (e.g. [Ziko et al., ICML'20]), optimal transport (e.g. [Lazarou et al., ICCV'21]), feature transformations (e.g. [Liu et al., ECCV'20]), information maximization (e.g [Veilleux et al., NeurIPS'21]), and transductive batch normalization (e.g [Bronskill et al., ICML'20]), among other works. We concede, however, that a related-work section, in which we move this introduction discussion and further augment it with more details on the objectives/differences, will be good. We propose to add the following paragraph to the paper:
>
> **Additional discussions on transductive few-shot objectives:**
> *Unlike most of the existing transductive few-shot methods, our MDL formulation does not encode strong assumptions on the label statistics of the query set (e.g. class balance or/and a perfect match between the query and support classes). Information maximization methods, such as [Veilleux et al., NeurIPS'21, Boudiaf et al., NeurIPS'20], maximise the confidence of the sample-wise predictions while encouraging class balance. Optimal transport methods [Lazarou et al., ICCV'21, Hu et al., ICANN'21] estimate an optimal mapping matrix, which could be viewed as a joint probability distribution over the features and the labels while imposing a hard class-balance constraint via the Sinkhorn-Knopp algorithm. Both information-maximization and optimal-transport methods have inherent class-balance bias and, as shown in the experiments, undergo a drastic drop in accuracy as the number of possible classes increases (as this corresponds to highly imbalanced problems). Prototype rectification [Liu et al., ECCV'20] transforms the query features so as to minimize the difference between the overall statistics of the query and support sets. This relies on the assumption that the support and query classes match perfectly, and explains the drastic drop in the performance of [Liu et al., ECCV'20] in our setting. Inspired by graphical models, the Laplacian regularization in [Ziko et al., ICML'20] is a pairwise label correction (rather than a learning) method, which encourages assigning the same label to query samples that are close in the input space; it does not learn a class representation from the query set (the prototypes being fixed as those of the support set) and therefore does not suffer from class-balance bias.*
>
> -------------
>
> **Q:** More details on the task generation procedure.
>
> **A:** Yes, the classes in the query set partially overlap with those in the support set. More precisely, for each few-shot task, we build the support and query set as follows:
>
> *Support:* For each of the $K$ possible classes of the test set (e.g. $K=20$ for mini), we randomly sample $n$ images according to the uniform distribution, where $n$ is the number of shots.
>
> *Query:* We first randomly pick $K_{\text{eff}} < K$ classes among the $K$ possible classes. We then randomly sample $|\mathbb{Q}|$ images among those belonging to the $K_{\text{eff}}$ classes but images that
> do not appear in the support set, with $|\mathbb{Q}|=75$. To clarify this in the paper, we will add the graphical illustration in Fig. A (right side) in the anonymous link we provide above (answer to all reviewers).
> The class color coding in the figure indicates whether a support class appears in the query set or not.
>
> Please note that, while the task-generation paragraph (second paragraph in section 5.1 in the experiments) is brief, we provided more details on task generation in Section 2 (last paragraph). We realize, however, that it is better to concentrate the information in these two paragraphs in the same experimental section, for better readability. We will do so.

---

> > ### Author Response · Authors · 2022-08-02
> > **Answer to reviewer #7GFu (Part 2)**
> >
> > **Q:** Fig. 1: Why the accuracy of some baseline methods actually increases with $K_{\text{eff}}$?
> >
> > **A:** Thanks for noticing this! Indeed, BD-CSPN [Liu et al., ECCV'20] is the only baseline whose performance monotonically (and
> > slightly) increases with $K_{\text{eff}}$. The performance of this method is also among the most affected when the class-overlap between the query and support sets decreases, i.e., when $K_{\text{eff}}$ gets smaller (left side of the plots) and/or $K$ gets bigger (e.g. tiered-ImageNet, with $K$ = 160). We hypothesize that this behavior is due to the following technical fact: BD-CSPN transforms the query features so as to minimize the shift between the overall statistics of the query and support sets (Eqs. (7) and (8) in [Liu et al., ECCV'20]). This encodes the prior that the support and query classes match perfectly. When $K_{\text{eff}}$ increases with $K$ fixed (left side of the plots), the overlap between the query and support classes of the tasks increases, which corresponds better to this prior and might explain the increase in the performance of BD-CSPN. We will discuss this in both related works and result interpretation sections as the transformation in [Liu et al., ECCV'20] is quite used in the recent transductive few-shot literature.
> >
> > ----------
> >
> > **Q:** Empirical evidence supporting the choice $\lambda = |\mathbb{Q}|$.
> >
> > **A:**  We provide additional experiments supporting our theoretical analysis in Figs. C and D, within the anonymous link above (answer to all reviewers). Fig. C plots the accuracy as a function of $\lambda$ for different values of $K_{\text{eff}}$, showing quite stable performance and confirming that the optimal empirical value does not deviate significantly from the theoretical one (i.e. $\lambda=|\mathbb{Q}|$). Fig.~D shows an approximately linear behavior of the optimal empirical $\lambda$ w.r.t $|\mathbb{Q}|$, which also supports our analysis.
> > As one could expect, when the class-imbalance level is high ($K_{\text{eff}}=2$), the optimal empirical value of $\lambda$ is higher than its theoretical counterpart. On the contrary, when the query set is better balanced ($K_{\text{eff}}=10$), the optimal value is under the theoretical one. Nonetheless, the gap in performance between the theoretical and empirical values remains small.
> >
> > --------
> >
> > **Q:** Experiments showing the limitations of PADDLE in a more balanced setting.
> >
> > **A:** In fact, the smaller the gap between $K_{\text{eff}}$ (effective classes) and $K$ (all possible classes), the more balanced the query sets. This setting corresponds to the right sides of the plots we provided in the first raw of Fig. 1 in the paper (miniImageNet; $K=20$; $K_{\text{eff}}$ varies on the x-axis), which shows that PADDLE's performance drops below TIM and $\alpha$-TIM as $K_{\text{eff}}$ approaches $K$. This makes sense as TIM and $\alpha$-TIM encode a prior on class balance. For tiered and i-Nat, we have a relatively more drastic form of class imbalance than on mini, as the number of possible classes $K$ is larger ($K=160$ for tiered and $K=227$ for iNat).
> >
> > --------
> > **Q:** On the link to existing MDL-based clustering algorithms.
> >
> > **A:** In the section on the connection to MDL (page 4), we discussed relation to the discrete label count in Eq. (3), which is often used in unsupervised clustering problems (e.g. [Delong et al., IJCV'12]).
> > In fact, our partition-complexity term could be viewed as an interesting continuous relaxation of the discrete label count in Eq. (3). To further clarify this, we provide in Figure D in the anonymous link above a graphical illustration of this connection in the case $K=2$. We also discussed the optimization difficulty ensuing from discrete label count (3), which is often tackled with combinatorial algorithms or cluster-merging heuristics, both of which are computationally intensive. While it also penalizes the number of effective classes, our continuous relaxation has several advantages over the discrete label count in Eq. (3), despite the surprising fact that it is not common in the MDL-based clustering literature, to the best of our knowledge. First, it is a continuous function of the assignment variables, which enabled us to derive a fast and converging primal-dual block coordinate descent algorithm. Second, it could be interpreted as a number of bits required to encode the set of classes, given class probabilities.

---

### Official Review · Reviewer_cgRh · 2022-07-14

**Rating:** 5
**Confidence:** 2
**Soundness:** 2 fair
**Presentation:** 2 fair
**Contribution:** 2 fair

**Summary:**

The paper introduces a Primal Dual Minimum Description Length (PADDLE) formulation, which balances data fitting accuracy and model complexity for a given few-shot task. It extends the current benchmarks, so that the query-set classes of a given task are unknown and do not match exactly the support-set classes, but just belong to a much larger set of possible classes. With a drop in performance on some of the best performing SOTA transductive few-shot methods, the paper introduces a hyper-parameter free MDL-inference formulation which balances data-fitting accuracy and model complexity for a given few-shot task, subject to supervision constraints from the support set.

**Questions:**

I think a few important baselines are missing. Please find then below:

1. Li C, Kothawade S, Chen F, Iyer R. PLATINUM: Semi-Supervised Model Agnostic Meta-Learning using Submodular Mutual Information. arXiv preprint arXiv:2201.12928. 2022 Jan 30.

2. Li, X., Sun, Q., Liu, Y., Zhou, Q., Zheng, S., Chua, T.-S., and Schiele, B. Learning to self-train for semi-supervised few-shot classification. Advances in Neural Information Processing Systems, 32:10276–10286, 2019.

**Limitations:**

See sections above.

**Strengths And Weaknesses:**

**Strengths**

- Block coordinate descent algorithm guarantees a convergence
- No hyperparameters need of tuning them
- The formulation can be used on top of any base class

**Weaknesses**

- The paper is hard to follow and is very dense. It would be great to add more empirical results to make it stronger.

---

> ### Author Response · Authors · 2022-08-02
> **Answer to reviewer #cgRh**
>
> **Q:**  More empirical results will make the paper stronger.
>
> **A:** We added evaluations of three transductive few-shot baselines suggested by Reviewer #m7wN (FEAT [Ye et al., CVPR’20], SIB [Yu et al., ICLR’20], DPGN [Yang et al., CVPR’20]).
> Please refer to the Table below in our answer to Reviewer #m7wN. Consistently with the initial observations made in Table 1 in the paper, we observe drops in the performances of these three additional baselines in our new setting.
>
> We would also like to kindly point to additional results that we provide in Figs. C and D within the anonymous link above (top-level answer to all reviewers). The plots in Figs. C and D report additional experiments supporting our theoretical analysis in Section 3, which prescribes setting $\lambda$ automatically to $|\mathbb{Q}|$. Fig. C plots the accuracy as a function of $\lambda$ for different values of $K_{\text{eff}}$, showing a quite stable performance w.r.t $\lambda$ and confirming that the optimal empirical $\lambda$ does not deviate significantly from the theoretical $\lambda$ (i.e. $\lambda=|\mathbb{Q}|$). Fig. D shows an approximately linear behavior of the optimal empirical $\lambda$ (i.e. the one with which the best accuracy is reached) w.r.t $|\mathbb{Q}|$, which also supports our analysis.
>
> We agree with the reviewer that additional experiments could only strengthen the paper.
> However, as mentioned by reviewers #7GFu and #ig28, we believe that the experiments are quite thorough, with i) evaluations of 7 state-of-the-art methods on three different benchmarks over a range of values of $K_{\text{eff}}~$ (Figs. 1 and 2); ii) sensible ablations demonstrating the utility of the partition complexity-term (Table 2) and the effectiveness of the proposed block coordinate descent algorithm (Fig. 3); iii) experiments supporting our theoretical analysis prescribing to set $\lambda$ automatically to $|\mathbb{Q}|$ (Figs. C and D).
>
> -----------
>
> **Q:** Important baselines (PLATINUIM [Li et al., ICML’22] and LST [Li et al., NeurIPS’19]) are missing in the evaluation.
>
> **A:** Thanks for pointing to PLATINUIM [Li et al., ICML’22]  and LST [Li et al., NeurIPS’19] baselines. These are, in fact, semi-supervised few-shot learning methods, which use extra unlabeled data during training. They belong to a different sub-category of methods in the few-shot literature and are, therefore, not directly comparable to transductive few-shot methods. In the footnote on page 2 of the paper, we mentioned briefly semi-supervised few-shot learning, how it is different from the popular transductive few-shot setting that we address in our work, and cited a well-known paper in this category ([Ren et al., ICLR’18]). We will add to this footnote citations to the references suggested by the reviewer (PLATINUIM and LST).

---

> > ### Comment · Reviewer_cgRh · 2022-08-08
> > **Thank you for your response**
> >
> > I have no other concerns. I would keep my score.

---

### Author Response · Authors · 2022-08-02
**Top-level answer (to all reviewers)**

We greatly appreciate the reviewers' insightful/constructive comments, and we are pleased that all four reviewers voted towards acceptance.
We are also glad that reviewers found our new few-shot setting timely/realistic (Reviewer #7GFu), practical/of positive impact on many related works (Reviewer #ig28), and very interesting (Reviewer #m7wN). Reviewers also pointed out the interest of our block-coordinate descent algorithm and MDL-based formulation (convergence guarantee, hyper-parameter free).

Below we address all the questions raised by the reviewers. In particular, we provide additional baseline evaluations and clarifications as requested. For the reviewers' convenience, we provide additional plots/illustrations/results in the following anonymous link (as we will refer to these in our answers below):

https://github.com/AnonymousForNeurIPS2022/PADDLE.git

---

### Meta-Review · Area_Chair_3drP · 2022-08-26

**Recommendation:** Accept
**Confidence:** Less certain

**Metareview:**

The reviewers have raised some concerns on the baselines used in the empirical evaluation. However, the additional experiments by the authors seem to have convinced the reviewers. Please make sure to add these additional experiments in the camera-ready version of the paper.

**Award:**

No

---

### Decision · Program_Chairs · 2022-09-14

Accept